# Young, but not in the dark—The influence of reduced lighting on gait stability in middle-aged adults

**Tirtsa Naaman, Roee Hayek, Itai Gutman, Shmuel Springer**[ID]*

Faculty of Health Sciences, Department of Physical Therapy, The Neuromuscular & Human Performance Laboratory, Ariel University, Ariel, Israel

* shmuels@ariel.ac.il

## Abstract

### Background

The aim of this study was to investigate the effects of walking in reduced lighting with or without performing a secondary cognitive task on gait dynamics in middle-aged adults and to compare them with young and old adults.

### Methods

Twenty young (age 28.8±4.1), 20 middle-aged (age 50.2±4.4), and 19 elderly (age 70.7 ±4.2) subjects participated in the study. Subjects walked on an instrumented treadmill at a self-determined pace under four conditions in randomized order: (1) walking in usual lighting (1000 lumens); (2) walking in near-darkness (5 lumens); (3) walking in usual lighting with a serial-7 subtraction dual-task; and (4) walking in near-darkness with a serial-7 subtraction dual-task. Variability in stride time and variability in the trajectory of the center of pressure in the sagittal and frontal planes (anterior/posterior and lateral variability) were measured. Repeated measures ANOVA and planned comparisons were used to determine the effects of age, lighting conditions, and cognitive task on each gait outcome.

### Results

Under usual lighting, stride time variability and anterior/posterior variability of the middle-aged subjects were similar to those of the young and lower than those of the old. The lateral variability of the middle-aged subjects was higher than that of young adults under both lighting conditions. Similar to the older adults, the middle-aged participants increased their stride time variability when walking in near-darkness, but they were the only ones to exhibit increased lateral variability and anterior/posterior variability in near-darkness. Young adult gait was not affected by lighting, and concurrent performance of a cognitive task while walking did not affect gait stability in all groups under any of the lighting conditions.

**Data Availability Statement:** The data underlying the results presented in this study are publicly available from the Zenodo repository (https://doi.org/10.5281/zenodo.7788344).

**Funding:** The author(s) received no specific funding for this work.

**Competing interests:** The authors have declared that no competing interests exist.

## Conclusions

Gait stability decreases in middle age when walking in the dark. Recognition of functional deficits in middle age could promote appropriate interventions to optimize aging and reduce fall risk.

## Introduction

Falls resulting from deterioration of gait stability are a major public health problem. Stable human locomotion requires coordination between somatosensory input and visual information that governs foot position control. The important role of vision in gait regulation has been well described [1]. Online visual information is critical for successfully negotiating obstacles or walking over complex terrain [2–5], and a sudden transition from normal to marginal illumination challenges movement control [6]. Reliance on visual information to maintain postural stability increases with age, and impaired visual perception due to a reduction in light is associated with high fall risk [7–9]. The ability to walk in reduced lighting conditions during aging is substantial for safe navigation in everyday environments and may help prevent falls; therefore, it may affect quality of life [10].

Most studies that have examined the effects of limited light intensity on walking have tested the variability of gait pattern, as it represents the ability to optimally control gait from one stride to the next and is considered a reliable way to quantify balance during locomotion [11]. Figueiro et al. [12] reported increased step length variability when older adults walked on a dim walkway. Kesler and colleagues [13] demonstrated that older adults with high levels gait disorders exhibit increased stride and swing time variability when lighting was reduced, and even healthy older adults experienced a slowing of gait speed. In healthy young adults, deprivation of visual information during treadmill walking increased step width variability, but stride time and stride length increased only when subjects walked slower than their preferred walking speed [14]. Another study that investigated the effect of visual deprivation (i.e., closed eyes) on dynamic stability showed no age-related differences in gait variability between young and older adults while walking on a treadmill [15]. Other measures of gait dynamics during treadmill walking, such as frontal plane trunk acceleration variability, did not differ when healthy adults aged 44±14 years walked with open or closed eyes [16].

Cohort-based population studies show that the annual prevalence of falls triples from 9% in 40- to 44-year-old adults to 28% in 60- to 64-year-old adults [17,18] suggesting that falls are not only a problem of old age. Gait variability has also been shown to increase at an accelerated rate at middle age, which may be a relevant early indicator of fall risk [19–21]. These results support the premise that middle age might be a critical life stage for identifying indicators for fall risk. It has been previously shown that increasing gait complexity may help to identify changes in balance in middle-aged adults [22].

Despite the increased prevalence of falls in middle age, the effects of reduced visual input on gait stability in middle-aged adults have rarely been investigated. Furthermore, many of the currently available references examined the effects of visual deprivation on dynamic stability while walking with eyes closed [14–16]. However, it has been shown that walking with eyes closed can be easier than walking with eyes open without visual cues [23]. Finally, assessing gait variability under challenging situations such as walking while performing a secondary cognitive task can also provide information about postural control during walking. Many studies have demonstrated a significant interaction between cognitive domains and walking control

[24–26]. In addition, the ability to walk while performing other tasks (e.g., talking on a smartphone while walking) is important for maintaining many daily activities [27].

Therefore, the aim of this study was to investigate the effects of reduced lighting during walking with or without a dual task on gait variability in middle-aged individuals and to compare their response with that of young and old adults. To better describe our cohort, we also examined physical performance and activity, as these have been associated with general health and cognitive function [28,29].

## Methods

### Participants

The study included a convenience sample of 20 young adults (age 28.8±4.1), 20 middle aged adults (age 50.2±4.4), and 19 older adults (age70.7±4.2).

Participants were included if they lived in the community, were independent in activities of daily living, and could walk without assistance. Subjects with neurological, orthopedic, vestibular, or visual impairments (e.g., age-related macular degeneration, glaucoma, cataract, diabetic retinopathy) or other comorbidities that could affect gait were excluded. The study was approved by Ariel University Ethics Committee (approval number AU-HEA-SS-20210809). All subjects gave written informed consent to participate in the study.

### Procedure

Each subject participated in a single session that lasted approximately 60 minutes.

Before assessing gait, anthropometric measures were collected, physical performance was quantified using handgrip strength (Jamar®, 5030J1, Patterson Medical, Warrenville, IL, USA) and the 30-second chair-stand test (30CST) [30], and physical activity was measured using the Beacke Physical Activity Questionnaire (BPAQ) [31].

Gait was assessed while the subjects walk over an instrumented treadmill (Zebris FDM-T; Zebris Medical GmbH, Isny, Germany) in a self-paced speed under four conditions in randomized order: (1) walking in usual lighting (1000 lumens); (2) walking in near-darkness (5 lumens); (3) walking in usual lighting with dual-task; and (4) walking in near-darkness with dual-task. Each gait trail lasted 3 min, with 3 min rest between tests. During the dual-task conditions, subjects walked while reciting out loud serial subtractions of 7, starting from a different 3-digit number at each trial. No instructions regarding priority of walking vs. cognitive task were given. Before performing the dual task while walking, the task was measured for 120 s while sitting, to examine the ability of the subjects to perform an arithmetic task the score of this test was based on number of correct responses (Serial-7 sitting). In addition, subjects were given the opportunity to become accustomed to walking on a treadmill in usual lighting for 6–7 minutes [32]. During the familiarization period gait speed was gradually increased and the maximum comfortable walking speed achieved by each subject was used for all gait tests. For safety reasons, subjects wore a harness (without body weight support) that did not impede movement of the arms and legs.

The Zebris instrumented treadmill consists of a capacitance-based foot pressure platform in the treadmill with a sensing area of $101.6 \times 47.4$ cm, containing 6720 sensors measuring at 240 Hz. The treadmill's dedicated software (FDM V1.18.48) provides data on spatiotemporal gait parameters and center of pressure (COP) trajectories during walking. The following gait outcomes were derived from the treadmill software [33]: (1). Stride time variability: the standard deviation of stride times divided by the average stride time; (2). Anterior/posterior variability: the standard deviation of the intersection point of the CoP trajectory on the frontal axis; (3). Lateral variability: the standard deviation of the intersection point of the CoP

trajectory on the transverse axis. The anterior/posterior and lateral variability were derived from the dedicated treadmill software. The software generates a graphical pattern representing a continuous trace of the CoP trajectory during walking, with "zero' deviations from the CoP intercepts corresponding to constant strides.

## Statistical analysis

Shapiro-Wilk test verified that all variables were normally distributed. Background variables (i.e., height, weight, serial-7 sitting, hand grip strength, 30CST, BPAQ, and treadmill walking speed) were compared between groups using ANOVA followed by post hoc analyses. A 3 X 2 X 2 repeated measure ANOVA and planned comparisons were used to determine the effect age group (young, middle age, Old) X lighting condition (usual lighting, near-darkness) X dual task (with/without dual task) on each gait outcome. An additional ANOVA examined the effect of age group on serial-7 task performance during walking (serial-7 walking). Partial η2 effect sizes were calculated for each ANOVA model, with η2 = 0.01 indicating a small effect, η2 = 0.06 indicating a medium effect, and η2 = 0.14 indicating a large effect [34]. Statistical analysis was performed using IBM SPSS Statistics, version 27.0. (Armonk, NY: IBM Corp), and significance was set at $p < 0.05$.

## Results

### Background variables

*Table 1* summarizes the background characteristics of the participants. There were significant differences between groups in maximal grip strength (p = 0.002, partial η2 = 0.182), ability to move from sitting to standing (p<0.001, partial η2 = 0.347), and self-selected treadmill walking speed (p = 0.003, partial η2 = 0.185). To test the possibility that these variables are covariates in the main analysis, we analyzed the correlation between all three variables and age. The correlations of each variable with age were larger and more significant than all other correlations. Therefore, age was considered as a mediating factor between these three variables and the dependent variables (i.e., gait outcomes). The older adults had slower self-selected treadmill

**Table 1. Participant characteristics.**

|  | Young (N = 20) | Middle age (N = 20) | Old (N = 19) | p-value |
|---|---|---|---|---|
| **Gender: Female (%)** | 10 (50%) | 10 (50%) | 10 (52%) | 0.986 |
| **Height (cm)** | 171.5±11.7 | 170.1±11.1 | 168.4±7.8 | 0.645 |
| **Weight (Kg)** | 70.4±15.3 | 68.6±15.4 | 70.3±10.5 | 0.905 |
| **BMI** | 23.7±3.4 | 23.4±3.4 | 24.8±3.3 | 0.425 |
| **BPAQ** | 7.3±2.4 | 6.8 ±1.7 | 6.2 ±1.3 | 0.213 |
| **Serial-7 sitting** | 21.4±9.2 | 19.6±10.9 | 14.2±8.8 | 0.064 |
| **Maximal grip[a] (KgF/Kg)** | 0.56±0.14 | 0.53±0.12 | 0.41±0.15*[#] | 0.002 |
| **CST30 (N)** | 27.6 ±6.7[#] | 21.6 ±5.6* | 17.6 ±4.7* | <0.001 |
| **Treadmill walking speed[b] (m/sec)** | 0.63±0.1 | 0.60±0.13 | 0.50±0.11*[#] | 0.003 |

Values are presented as mean ±SD.

BMI:Body mass index; CTS30:Chair-stand test 30 seconds; BPAQ:Beacke physical activity questionnaire.

[a] normalized to body weight

[b] normalized to height.

*significant difference compared to young.

[#]significant difference compared to middle age.

walking speed and lower grip strength compared to the middle-aged (p = 0.004, and p = 0.018 respectively) and the young adults (p = 0.028, and p = 0.005, respectively); however, there were no significant differences in these parameters between the middle-aged and young groups. The young adults had a better ability to move from sitting to standing than the middle-aged adults (p = 0.005) and old adults (p<0.001), while there was no significant difference between the middle-aged and old groups.

## Gait variability

The gait variability results can be found in Appendix 1 in S1 File.

**Stride time variability.**   A significant effect on stride time variability was found for age (p<0.001, partial $\eta^2$ = 0.482) and lighting condition (p<0.001, partial $\eta^2$ = 0.523), but not for dual-task (p = 0.289). In addition, a significant interaction was found between the effects of age and lighting condition (p = 0.018, partial $\eta^2$ = 0.128).

The older group had significantly higher stride time variability than the middle-aged and young groups (p<0.001 in both comparisons), but there was no difference in stride time variability between the middle-aged group and the young group (p = 0.441). The difference in stride time variability between the old and middle-aged groups was smaller in near-darkness than in usual lighting.

Within-group comparisons of stride time variability in the three groups under the two lighting conditions is shown in Fig 1. The old and middle-aged groups increased their stride time variability in near-dark compared to usual lighting (p<0.001 in both comparisons, partial $\eta^2$ = 0.496 and 0.732 respectively), whereas the young adults' stride time variability did not change (p = 0.069).

**Anterior/Posterior variability.**   A significant effect on anterior/posterior variability was found for age (p<0.001, partial $\eta^2$ = 0.319) and lighting condition (p = 0.007, partial $\eta^2$ = 0.143), but not for dual-task (p = 0.561). There was no interaction between the different effects. To further investigate the effect of near darkness, our between-groups analysis showed a

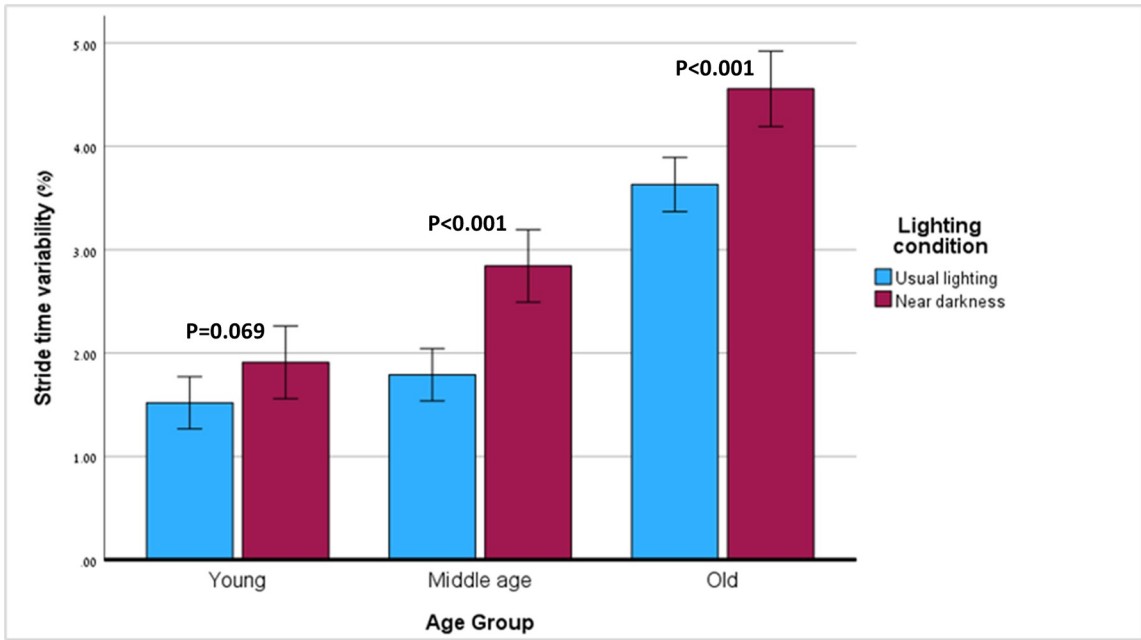

**Fig 1. Stride time variability in the two lighting conditions—within-group comparisons.**

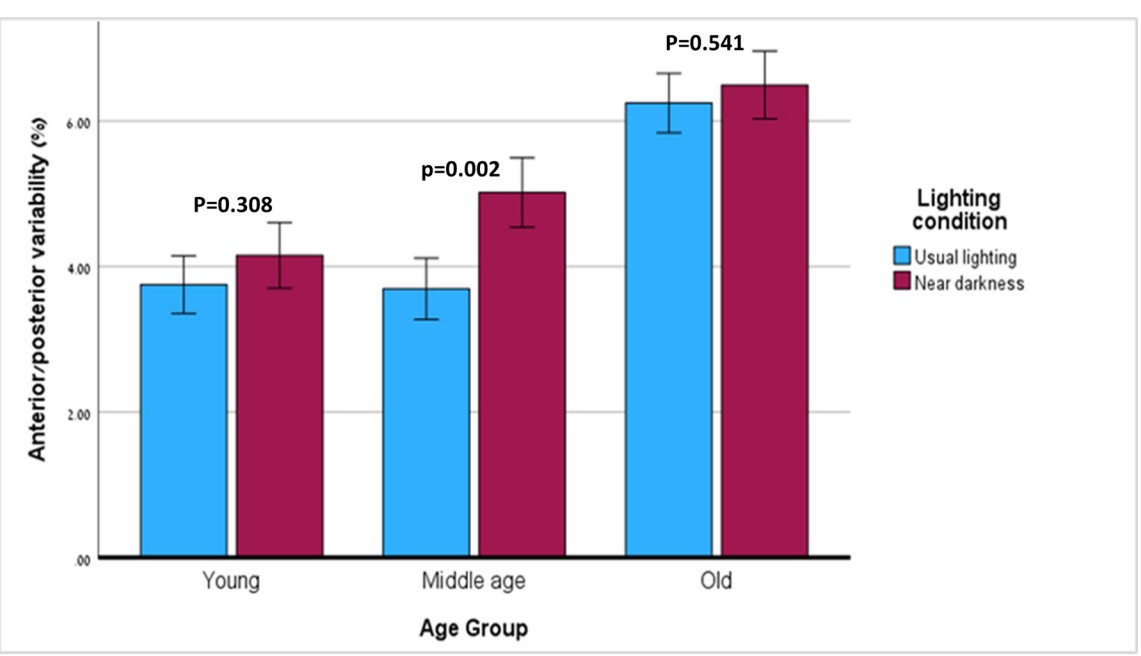

**Fig 2. Anterior/Posterior variability in the two lighting conditions—within-group comparisons.**

significant difference in anterior/posterior variability between the old group and the young group (6.37, 95% CI: 5.59–7.15 vs. 3.953, 95% CI: 3.194–4.712, p<0.001) and between the old group and the middle-aged group (6.37 vs. 4.36, 95% CI: 3.55–5.16, p = 0.002), but not between the middle-aged group and the young group (p = 1.000).

Within-group comparisons of anterior/posterior variability in the three groups under the two lighting conditions is shown in Fig 2. The middle-aged group had increased anterior/posterior variability in near-dark compared to usual lighting (3.69, 95% CI: 2.85–4.54 in usual light versus 5.02, 95% CI: 4.05–5.98in near dark, p = 0.002, partial $\eta^2$ = 0.657). In contrast, the young and the old groups did not change their anterior/posterior variability (p = 0.308, p = 0.541, respectively).

**Lateral variability.** A significant effect on lateral variability was found for age (p<0.001, partial $\eta^2$ = 0.427) and lighting condition (p = 0.004, partial $\eta^2$ = 0.157), but not for dual-task (p = 0.204). Also, a significant interaction was demonstrated between the effects of age group and lighting condition (p = 0.009, partial $\eta^2$ = 0.177). The old group had increased lateral variability compared to the middle-aged group (11.58 vs. 7.11, 95% CI: 5.21–9.02, p = 0.005) and the young group (11.58, 95% CI: 9.67–133.48 vs. 3.59, 95% CI: 1.74–5.44, p<0.001). The middle-aged group had increased lateral variability compared to the young group (p = 0.031).

Within-group comparisons of lateral variability in the three groups under the two lighting conditions is shown in Fig 3. The middle-aged group increased their lateral variability in near-dark compared to usual lighting (6.22, 95% CI: 4.43–8.00 in light versus 8.01, 95% CI: 5.90–10.11 in near-dark, p<0.001, partial $\eta^2$ = 0.505), while the young and the old groups did not change their lateral variability (p = 0.247 and p = 0.832 respectively).

**Serial -7 walking.** There were significant differences between groups in serial-7 performance during walking (p = 0.042, partial $\eta2$ = 0.111). The older adults had fewer correct responses compared to the young groups (10.4, 95% CI: 7.2–13.6 vs. 15.3, 95% CI: 13.2–17.5, p = 0.042) but not compared to the middle-aged group (13.8, 95% CI: 10.6–17.0, p = 0.262), and there was no significant difference in serial-7 walking between the middle-aged and young groups (p = 0.262).

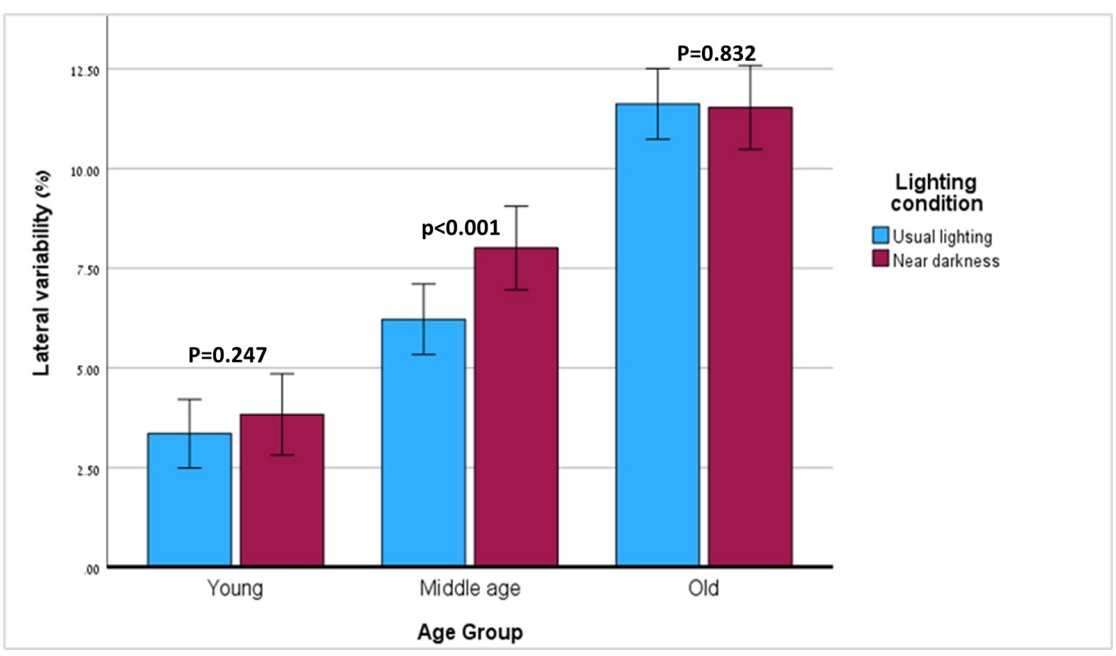

**Fig 3. Lateral variability in the two lighting conditions—within-group comparisons.**

## Discussion

The results of the present study show that middle-aged subjects were more affected by the reduced light condition compared with young and old adults. Similar to old adults, middle-aged participants increased their stride time variability when walking in near-darkness, but were the only ones who increased lateral variability and anterior/posterior variability in near-darkness, whereas gait variability of young adults was not affected by the light condition. In addition, middle-aged individuals showed increased lateral variability in both lighting conditions compared with young adults.

Our findings may suggest that age-related changes beginning in middle age impair the ability to regulate gait stability when visual perception is limited due to diminished lighting. Decreased ability of the somatosensory and vestibular systems to compensate for visual input has been suggested as a reason for slow walking speed and decreased postural instability under limited lighting conditions in the elderly population [8,35]. This could also be a possible cause for the increased gait variability observed in the middle-aged group in our study. Indeed, there is evidence that deficits in proprioception or vestibular function may begin in middle age. Hurley et al. [36] and Wingert et al. [37] examined the acuity of joint position sense in young, middle-aged, and elderly subjects and reported age-related deterioration in proprioception. Age-related loss of vestibular function is also common in middle age. A national survey of U.S. adults aged 40 years and older found that the prevalence of vestibular dysfunction was 18.5% in adults aged 40–49 years and increased to 33.0% in adults aged 50–59 years [38,39]. Another possible explanation for the decreased balance during walking in the middle-aged group might be related to the results of the 30CST. While our subjects in all three groups had no walking impairments and were physically active according to their BPAQ score [31,40], the 30CST score in the middle-aged group was below the previously reported age reference value [41] and were similar to those in the old group. Reduced ability to move from sitting to standing has been associated with decreased dynamic stability [42,43]. Therefore, the low 30CST score

in our middle-aged group may reflect age-related changes that could affect balance and gait. Overall, human performance in middle age is highly variable [44]. Thus, further research should be conducted to confirm our findings and proposed explanations.

In contrast to our findings, Reynard and Terrier [16] examined gait variability during blindfolded treadmill walking in healthy adults with a mean age of 44±14 years and reported no destabilizing effects on gait. Several explanations can be proposed for this difference. Our subjects were tested with their eyes open in a near-dark environment. It has been shown that it may be more difficult to control gait when walking with eyes open in the dark than when walking with eyes closed (Yelnik et al. 2015). In addition, our subjects were tested under both lighting conditions at their self-selected gait speed measured in normal light, whereas Reynard and Terrier compared gait variability between walking conditions with blindfolded gait speed much slower than normal walking speed. This slower and more careful walking may have allowed subjects to use alternative sensory strategies to control gait dynamics without the aid of vision, so that they did not exhibit impaired stability. Indeed, it has been previously observed that healthy subjects who had been accustomed to reduced light conditions for a sufficiently long time tended to have a near-normal gait (Moe-Nilssen et al. 2006). However, in everyday walking scenarios, there is often a sudden transition from normal to low light or near-darkness, such that the habituation time required to develop a careful gait is insufficient to reduce the challenge of motion control. Finally, while we focused on the middle age, the ages of Reynard and Terrier subjects varied, and their mean age was<45 years. The age difference between subjects and the methods and technologies used to record and analyze gait variability could also explain the different results.

Another study that may not agree with our findings is the study by Kesler et al. [13], which reported that walking in reduced lighting had no effect on the variability of stride and swing time in healthy older people. While Kesler et al. examined overground walking, it is possible that the inherent balance challenges associated with treadmill walking [45] amplified the effects of reduced lighting on stride time variability in the middle-aged and older subjects in our study. In addition, various measures of variability should be measured to assess gait stability [46]. Our study, which assessed both temporal and spatial aspects of gait variability, may improve the understanding of the effects of reduced lighting on gait dynamics. While the middle-aged participants showed increased variability in all three gait outcomes in near darkness, the older adult group did not change their spatial aspects of gait variability (i.e., lateral and anterior/posterior variability) in near darkness. A possible explanation could be related to the high lateral and anterior/posterior variability observed in the older adult group under usual lighting. It is possible that gait stability did not deteriorate further and was less sensitive to the effect of reduced lighting because of the already increased variability values. It should also be noted that our results are consistent with previous data showing higher COP gait variability in middle-aged individuals compared to young adults [20] and the importance of measuring mediolateral dynamic stability outcomes in middle age [19].

Performing an additional task simultaneously with walking did not affect gait stability in either age group under both normal and reduced light conditions. The literature on the effects of cognitive tasks on gait variability in healthy individuals is inconclusive. Some studies show a destabilizing effect, especially in the elderly [47,48], whereas other studies do not [24,49,50]. Our results are more consistent with the latter studies. The type and difficulty of the dual task may influence its effect on gait [51]. Although it has been suggested that walking in near-darkness requires more attention than walking in normal lighting [13], it is possible that the added complexity of walking in near-darkness with counting backward by 7 did not elicit sufficient cognitive load to alter gait stability. Further research is needed to understand why certain cognitive tasks may affect gait variability while other tasks do not.

Although stability during walking was not affected by the additional cognitive task, serial 7 performance of older adults was lower than that of young adults during walking, whereas it did not differ during sitting. This result is consistent with the "posture-first" principle, which has shown that older people prefer to maintain stability when walking while performing another task [52].

This study has several limitations. Our middle-aged group included subjects aged 45 to 65 years. Although middle age is associated with this stage of life, human performance can vary widely during this period. We believe that the results of the present study shed light on the effects of reduced lighting on gait stability in middle age. Nevertheless, future studies with a larger and varied sample may allow analysis of age subgroups within middle age. Another limitation of our study is that it was conducted in an indoor laboratory setting while subjects walked on a treadmill wearing a safety harness. It has already been shown that a non-weight-supporting harness does not alter gait dynamics [53] and that the variability measured during treadmill walking may be an acceptable representation of walking on the ground [54]. However, it is also possible that the effect of reduced lighting would have been greater if subjects had been tested walking overground. Therefore, to raise the ecological validity, it is recommended that future studies also examine the effect of reduced lighting on gait stability in middle-aged adults in an outdoor environment. To further verify our results, it is recommended that such studies will include additional measures of gait stability. Finally, our subjects had no visual impairment or uncorrected problem related to visual acuity. Nevertheless, we did not directly examine aspects of vision as well as other factors, such as vestibular function, that may affect gait. It would also be helpful to investigate these and other relevant aspects in future studies.

## Conclusions

Apart from increased lateral CoP trajectory variability, gait stability in full illumination in middle-age is similar to that of younger people and differs from locomotion in old adults. However, when walking in the dark, a deterioration in balance is mostly pronounced in middle-aged walking. Performing an additional task simultaneously with walking did not affect gait stability in either age group in both normal and reduced light conditions. The results may highlight the need to further investigate postural control in middle-aged adults, as detection of functional deterioration in middle age could promote appropriate interventions that could support better and healthy aging.

## Supporting information

**S1 File. Appendix 1: Gait variability results.**
(DOCX)

## Acknowledgments

The authors would like to thank the participants in this study.

## Author Contributions

**Conceptualization:** Tirtsa Naaman, Shmuel Springer.

**Data curation:** Tirtsa Naaman.

**Formal analysis:** Tirtsa Naaman, Roee Hayek, Itai Gutman, Shmuel Springer.

**Investigation:** Shmuel Springer.

**Methodology:** Tirtsa Naaman, Roee Hayek, Shmuel Springer.

**Project administration:** Tirtsa Naaman, Shmuel Springer.

**Software:** Itai Gutman.

**Supervision:** Shmuel Springer.

**Validation:** Itai Gutman.

**Writing – original draft:** Tirtsa Naaman, Shmuel Springer.

**Writing – review & editing:** Tirtsa Naaman, Roee Hayek, Itai Gutman, Shmuel Springer.

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
