## [Decision Letter · Decision Letter 0]

23 Feb 2023

PONE-D-22-35579Young, but not in the Dark - the Influence of Reduced Lighting on Gait Stability in Middle-aged adultsPLOS ONE

Dear Dr. springer,

Thank you for submitting your manuscript to PLOS ONE. After careful consideration, we feel that it has merit but does not fully meet PLOS ONE’s publication criteria as it currently stands. Therefore, we invite you to submit a revised version of the manuscript that addresses the points raised during the review process.

We look forward to receiving your revised manuscript.

Kind regards,

Ryan Thomas Roemmich

Academic Editor

PLOS ONE

Journal Requirements:

Reviewers' comments:

Reviewer's Responses to Questions

**Comments to the Author**

1. Is the manuscript technically sound, and do the data support the conclusions?

Reviewer #1: Partly

Reviewer #2: Partly

2. Has the statistical analysis been performed appropriately and rigorously? 

Reviewer #1: No

Reviewer #2: No

3. Have the authors made all data underlying the findings in their manuscript fully available?

Reviewer #1: Yes

Reviewer #2: No

4. Is the manuscript presented in an intelligible fashion and written in standard English?

Reviewer #1: Yes

Reviewer #2: Yes

5. Review Comments to the Author

Reviewer #1: The authors present a study of the effects of dual task and environmental light levels on gait stability during treadmill walking in young, middle-aged, and older adults. I have presented a list of major and minor concerns below that can make the manuscript a better paper and more suitable for publication, however, it is my professional opinion that these data are not suitable for publication in PLOS One. The authors should consider publication in more gait and posture specific journals.

Reviewer #2: Introduction

Consider adding citations regarding the concept of increasing task challenge to identify changes in balance in middle-aged adults. (Changes in the control of obstacle crossing in middle age become evident as gait task difficulty increases, B.C. Muir, J.M. Haddad, R.E.A. van Emmerik, S. Rietdyk, https://doi.org/10.1016/j.gaitpost.2019.01.035)

The authors have not justified the following;

Why is walking in reduced lighting important/ interesting?

Why is a cognitive task important/ interesting?

Methods

Line 108 – “during 6-7 minutes”. Should be “for 6-7 minutes”.

How are the AP and ML variabilities actually calculated?

The authors only describe 3 variables in this analysis, other variables could be included.

What about other measures of stability including Step width, step length, trunk sway, etc?

Results

If there is a significant interaction the main effects are no longer valid. Please modify your results and discussion sections to reflect this. The interaction should be presented first, or indicate that there was no interaction. If there is an interaction describe the post hoc analysis, if not describe the main effects.

Figures – consider putting brackets between the columns compared. It is difficult to identify which comparisons are significant.

Anterior/posterior variability did not show any interactions effects, therefore you cannot discuss interaction effects in line 172-178 between age and lighting effects .

Figure 2 horizontal axis has a typo. Figure 2 should also be collapsed to reflect the effect of age across all lighting and cognitive conditions and the effects of lighting across all ages and cognitive conditions. This should be two graphs as there is no interaction. The current figure and text are misleading.

Where are the results of the cognitive task?

Discussion

You only presented three dependent variables with respect to light and age. OA changed in one, MA changed in 2. You can’t say anything about age and light interaction in AP variability. How can you say that “middle-aged subjects were most affected by the reduced light condition compared with young and old adults”

6. PLOS authors have the option to publish the peer review history of their article (what does this mean?). If published, this will include your full peer review and any attached files.

Reviewer #1: **Yes: **Katherine J Hunzinger

Reviewer #2: No

---

## [Author Response · Author response to Decision Letter 0]

31 Mar 2023

Response to the Comments of the Reviewers on Manuscript ID PONE-D-22-35579

We thank the reviewers for their comments and for the opportunity to consider a revised version of this manuscript. We revised the paper and attempted to incorporate the reviewers’ suggestions. Below, we summarize each of the comments raised by the reviewers and present our responses.

Reviewer #1: 

1. The authors present a study of the effects of dual task and environmental light levels on gait stability during treadmill walking in young, middle-aged, and older adults. I have presented a list of major and minor concerns below that can make the manuscript a better paper and more suitable for publication, however, it is my professional opinion that these data are not suitable for publication in PLOS One. The authors should consider publication in more gait and posture specific journals.

Response: We hope that our manuscript will meet the publication criteria of PLOS ONE after incorporating the suggestions of the reviewers. We believe that if our data should be considered for publication, there should be no difference between PLOS ONE and "gait and posture journals." PLOS ONE publishes many articles dealing with gait and the relationship between gait and aging. Please see- 

https://journals.plos.org/plosone/search?filterJournals=PLoSONE&q=gait&utm_content=a&utm_campaign=ENG-467

Methods:

2. Were subjects matched in any way (sex? Concussion history?) Authors should consider concussion history if possible in sensitivity anlayses

Response: Subjects were matched by sex, as shown in Table 1. It should also be noted that subjects did not differ in their BMI, level of physical activity, or ability to perform the serial-7 arithmetic task. Although concussion is a common sports injury ( particularly in contact sports), it is much less common and impaired in the general population. In a national health survey conducted by Health Canada, respondents were asked about the type of injury that was severe enough to limit their normal activities in the previous 12 months. According to this survey, the annual prevalence of Canadians reporting a concussion as their most severe injury was 110 per 100,000 population (Gordon et al. Pediatric neurology 2006). Furthermore, concussion does not appear to adversely affect or predict neurologic function in physically active individuals in early to middle adulthood, as the reviewer writes in an interesting article. Therefore, we believe that our paper should not include any analysis related to concussions.

3. Any concern that the dim lighting didn’t affect gait as much as hypothesized due to walking on the treadmill and not overground? Please elaborate in the limitations section.

Response: Was done as suggested.

4. Due to group differences in baseline demographics that may affect gait, it would behoove the authors to consider a sensitivity analysis comparing groups when controlling for these factors using an ANCOVA.

Response: We thank the reviewer for this comment. Following the reviewer's suggestion, we tested the possibility that these variables are covariates in the main analysis. To this end, we analyzed the correlation between all three background variables that differed between groups and age. This analysis revealed that the correlations of each variable with age were larger and more significant than all other correlations. Therefore, age was considered as a mediating factor between these three variables and the dependent variables (i.e., gait outcomes). This information was added to the revised manuscript.

Results/Discussion:

5. Failure to account for the below normal physical abilities of the middle-aged group may have biased your results and thus clinical interpretations of the data. I would reanalyze the data with control for the confounders, and re-write these sections accordingly.

Response: We assume that the reviewer is referring to the point we raised in the discussion that the 30CST score in our middle-aged group was below the age reference value reported earlier by Tveter et al, 2014. It should be clarified that although the total sample in the Tveter et al. study was 370, the sample sizes of the middle-aged subgroups were relatively small. For example, there were fewer than 30 women in the 50-59 age group (n=27). This sample is insufficient to indicate that our sample (n=20) had below-normal physical abilities. As mentioned in the discussion, human performance in middle age is highly variable, and further research should be conducted.

Minor Concerns

Abstract:

6. Try to use key words that aren’t in the title to increase searchability of the manuscript (e.g., older adults, walking, visual system, sensory reorganization)

Response: Thank you for this suggestion. The keywords have been changed.

7. Include effect sizes for results

Response: Because of the journal's restriction on the number of words in the abstract, effect sizes were presented only in the main text.

Methods:

8. L90-95: These assessments come out of nowhere, and therefore an argument for their inclusion should be made in the introduction, not in the methods section.

Response: Was done as suggested.

9. Include statement on effect sizes interpretations with citation

Response: Was done as suggested.

Results:

10. Very wordy, see comment below on Table 2 as a means to make this section more concise.

Response: The text has been shortened and Table 2 has been moved to the Supplementary Materials section. 

Tables:

11. Table 1: Include effect sizes

Response: Relevant effect sizes were included in the text. 

12. Table 2: Can likely add p-values/effect sizes here to eliminate the lengthy text in the results section. Also consider limiting to 2 decimal places

Response: Table 2 has been moved to the Supplementary Materials section (Appendix 1) and is now limited to 2 decimal places as suggested.

Figures:

13. Please reupload Figures 1-3 as they are blurry. It may benefit the reader as well if the authors show individual data points or a violin plot with boxplot overlay instead of a bar chart to better highlight individual data points.

Response: The figures were prepared according to the requirements of the journal and uploaded using the submission software. A violin plot with boxplot makes it easy to see the median and quartiles as well as the minimum and maximum values, but does not show mean values. Because the figures show a comparison between 3 groups under two lighting conditions, we believe that the bar graphs are more appropriate for visualizing the mean comparisons.

14. Figures 1-3 are Table 2 represented visually, pick one or the other as they are repetitive. My recommendation would be to keep the figures, remove the tables, but include the descriptions/numbers in the result section.

Response: Was done as suggested. Table 2 has been moved to Supplementary Materials section (Appendix 1). 

Discussion

15. Should add regarding lack of differences in the older adult group

Response: Thank you. We refer to this point in the revised discussion. 

Reviewer #2:

Introduction

1. Consider adding citations regarding the concept of increasing task challenge to identify changes in balance in middle-aged adults. (Changes in the control of obstacle crossing in middle age become evident as gait task difficulty increases, B.C. Muir, J.M. Haddad, R.E.A. van Emmerik, S. Rietdyk, https://doi.org/10.1016/j.gaitpost.2019.01.035)

Response: Thank you for this comment. This reference has been added as suggested.

2. The authors have not justified the following; Why is walking in reduced lighting important/ interesting? Why is a cognitive task important/ interesting?

Response: Thank you for raising these two important issues. According to the reviewer's comment, the importance of the ability to walk in reduced lighting as well as when performing cognitive tasks is explained in the revised manuscript.

Methods

3. Line 108 – “during 6-7 minutes”. Should be “for 6-7 minutes”.

Response: Was corrected. 

4. How are the AP and ML variabilities actually calculated?

Response: The anterior/posterior and lateral variability were derived from the dedicated treadmill software. The software generates a graphical pattern representing a continuous trace of the CoP trajectory during walking, where "zero' deviations from the CoP intersection points are equivalent to constant strides. This has been clarified in the revised manuscript. For more details, see also - Kalron and Frid, J Neurol Sci 2015. 

5. The authors only describe 3 variables in this analysis, other variables could be included. What about other measures of stability including Step width, step length, trunk sway, etc?

Response: We focused on gait variability outcomes to be consistent with most previous studies that have investigated the effects of limited lighting on walking and tested variability in gait patterns. Furthermore, in addition to step time variability, a reliable method for quantifying balance, our study also examined variability in COP pattern, a finding that is considered highly sensitive to differences between age groups (see Bizovska et al, Gait & Posture 2014). Nonetheless, based on the reviewer's comment, we add a sentence to the limitations section regarding the need to verify our results with additional measures of stability. 

Results

6. If there is a significant interaction the main effects are no longer valid. Please modify your results and discussion sections to reflect this. The interaction should be presented first, or indicate that there was no interaction. If there is an interaction describe the post hoc analysis, if not describe the main effects.

Response: We thank the reviewer for this comment and the opportunity to clarify this issue. When conducting an ANOVA, if there is a significant interaction effect between two independent variables (meaning that the effect of one variable on the dependent variable depends on the level of the other variable), this does not mean that the main effects are no longer valid. The main effects still represent the average effect of each independent variable on the dependent variable, ignoring the effects of the other independent variables. 

In fact, it is common for both main effects and interaction effects to be presented. In such cases, it is important to interpret both the main effects and the interaction effects to obtain a comprehensive understanding of the relationship between the independent variables and the dependent variable. 

Please also see -Tabachnick, B. G., & Fidell, L. S. (2019). Using multivariate statistics (7th ed.). Pearson. On page 385, the authors state that "a significant interaction indicates that the effect of one variable depends on the level of another variable. [...] However, a significant interaction does not render the main effects invalid. The main effect of each variable reflects the effect of that variable while controlling for the other variable(s)." Please see also our response to comment #8. 

7. Figures – consider putting brackets between the columns compared. It is difficult to identify which comparisons are significant.

Response: Based on the reviewer's comment, we have modified all 3 figures to show only our intended comparisons within the groups. We have also included in the figure the P value of each within-group comparison.

8. Anterior/posterior variability did not show any interactions effects, therefore you cannot discuss interaction effects in line 172-178 between age and lighting effects .

Response: We thank the reviewer for this comment and the opportunity to clarify this issue. There is a difference between planned comparisons and unplanned post hoc tests. Unlike an unplanned post hoc test, a planned comparison of a main effect can be conducted even if there is no interaction between the main effects in a factorial design. In such cases, the planned comparisons help to explore the nature and direction of the main effect. Please see- https://homepages.inf.ed.ac.uk/bwebb/statistics/Planned-comparison_post-hoc.pdf and also- Wei et al., Amino Acids, 2012. Following the reviewer’s comment, it was clarified in the revised manuscript that this was a planned post hoc comparison. 

9. Figure 2 horizontal axis has a typo. Figure 2 should also be collapsed to reflect the effect of age across all lighting and cognitive conditions and the effects of lighting across all ages and cognitive conditions. This should be two graphs as there is no interaction. The current figure and text are misleading.

Response: The typo has been corrected, thank you. Based on the reviewer's comment, we have changed all 3 figures to show only our intended comparisons within the groups, without referring to the interaction. 

10. Where are the results of the cognitive task?

Response: Based on the reviewer's comments, we have included the cognitive task data in the revised results and discussion.

11. Discussion- You only presented three dependent variables with respect to light and age. OA changed in one, MA changed in 2. You can’t say anything about age and light interaction in AP variability. How can you say that “middle-aged subjects were most affected by the reduced light condition compared with young and old adults”

Response: Similar to old adults, middle-aged participants increased their stride time variability when walking in near-darkness, but were the only group to increase lateral variability and anterior/posterior variability in near-darkness, whereas young adults gait variability was not affected by lighting conditions. Because old adults were affected by darkness only with respect to stride time variability and young adults were not affected at all, it is reasonable to conclude that middle-aged participants were more affected. Nevertheless, following this comment we replaced the word “most” with “more”. We also add a sentence to the limitations section regarding the need to verify our results with additional measures of stability.

---

## [Decision Letter · Decision Letter 1]

2 May 2023

PONE-D-22-35579R1Young, but not in the Dark - the Influence of Reduced Lighting on Gait Stability in Middle-aged adultsPLOS ONE

Dear Dr. springer,

Thank you for submitting your manuscript to PLOS ONE. After careful consideration, we feel that it has merit but does not fully meet PLOS ONE’s publication criteria as it currently stands. Therefore, we invite you to submit a revised version of the manuscript that addresses the points raised during the review process.

The reviewers have recommended that this paper be accepted pending the minor revisions suggested by Reviewer #1 below.

We look forward to receiving your revised manuscript.

Kind regards,

Ryan Thomas Roemmich

Academic Editor

PLOS ONE

Journal Requirements:

Reviewers' comments:

Reviewer's Responses to Questions

**Comments to the Author**

1. If the authors have adequately addressed your comments raised in a previous round of review and you feel that this manuscript is now acceptable for publication, you may indicate that here to bypass the “Comments to the Author” section, enter your conflict of interest statement in the “Confidential to Editor” section, and submit your "Accept" recommendation.

Reviewer #1: All comments have been addressed

Reviewer #2: All comments have been addressed

2. Is the manuscript technically sound, and do the data support the conclusions?

Reviewer #1: Yes

Reviewer #2: Yes

3. Has the statistical analysis been performed appropriately and rigorously? 

Reviewer #1: Yes

Reviewer #2: Yes

4. Have the authors made all data underlying the findings in their manuscript fully available?

Reviewer #1: Yes

Reviewer #2: No

5. Is the manuscript presented in an intelligible fashion and written in standard English?

Reviewer #1: Yes

Reviewer #2: Yes

6. Review Comments to the Author

Reviewer #1: Authors did a sufficient job incorporating reviewer concerns. My only concern is the removal of post-hoc and stating "planned comparisons" multiple times throughout the manuscripts make it seem like it wasn't planned due to the excessive nature of restating that it was planned (most things in research are already planned, so why specify so much?). Otherwise, happy with the submission!

Reviewer #2: My comments have been addressed. Thank you.

Adding additional required characters to meet the minimum.

7. PLOS authors have the option to publish the peer review history of their article (what does this mean?). If published, this will include your full peer review and any attached files.

Reviewer #1: **Yes: **Katie Hunzinger

Reviewer #2: No

---

## [Author Response · Author response to Decision Letter 1]

3 May 2023

Response to the Comments of the Reviewers on Manuscript ID PONE-D-22-35579R1

We thank the reviewers and the editor for the opportunity to consider a revised version of this manuscript. Both reviewers have indicated that we have addressed all of the comments and concerns they raised and have recommended that our paper be accepted, subject to the minor revision suggested by reviewer #1. We present that comment and our response below.

Reviewer #1: 

Authors did a sufficient job incorporating reviewer concerns. My only concern is the removal of post-hoc and stating "planned comparisons" multiple times throughout the manuscripts make it seem like it wasn't planned due to the excessive nature of restating that it was planned (most things in research are already planned, so why specify so much?). Otherwise, happy with the submission!

Response: The reviewer is right and we thank the reviewer for this comment. Based on this comment, we have reduced the multiple mentions of "planned comparisons" throughout the manuscript.

---

## [Editor Report · Decision Letter 2]

5 May 2023

Young, but not in the Dark - the Influence of Reduced Lighting on Gait Stability in Middle-aged adults

PONE-D-22-35579R2

Dear Dr. springer,

We’re pleased to inform you that your manuscript has been judged scientifically suitable for publication and will be formally accepted for publication once it meets all outstanding technical requirements.

Kind regards,

Ryan Thomas Roemmich

Academic Editor

PLOS ONE
---

## [Editor Report · Acceptance letter]

9 May 2023

PONE-D-22-35579R2 

Young, but not in the Dark - the Influence of Reduced Lighting on Gait Stability in Middle-aged adults 

Dear Dr. Springer:

I'm pleased to inform you that your manuscript has been deemed suitable for publication in PLOS ONE. Congratulations! Your manuscript is now with our production department. 

Kind regards, 

on behalf of

Dr. Ryan Thomas Roemmich 

Academic Editor

PLOS ONE